# Effects of Soil Temperature, Water Content, Species, and Fertilization on Soil Respiration in Bamboo Forest in Subtropical China

**Houxi Zhang [1], Zhuangzhuang Qian [2] and Shunyao Zhuang [2,*]**

[1] Forestry College, Fujian Agriculture and Forestry University, Fuzhou 350002, Fujian Province, China; zhanghouxi@fafu.edu.cn

[2] State Key Laboratory of Soil and Sustainable Agriculture, Institute of Soil Science, Chinese Academy of Sciences, 210008 Nanjing, Jiangsu Province, China; zzqian@njfu.edu.cn

[*] Correspondence: syzhuang@issas.ac.cn

**Abstract:** Understanding the change pattern of soil respiration (SR) and its drivers under different bamboo species and land management practices is critical for predicting soil $CO_2$ emission and evaluating the carbon budget of bamboo forest ecosystems. A 24-month field study was performed in subtropical China to monitor SR in experimental plots of local bamboo (*Phyllostachys glauca*) without fertilization (PG) and commercial bamboo (*Phyllostachys praecox*) with and without fertilization (PPF and PP, respectively). The SR rate and soil properties were measured on a monthly timescale. Results showed that the SR rate ranged from 0.38 to 8.53 μmol $CO_2$ m$^{-2}$s$^{-1}$, peaking in June. The PPF treatment had higher SR rates than the PP and PG treatments for most months; however, there were no significant differences among the treatments. The soil temperature (ST) in the surface layer (0–10 cm) was found to be the predominant factor controlling the temporal change pattern of the monthly SR rate in the PG and PP treatments (i.e., those without fertilization). A bivariate model is used to show that a natural factor—comprised of ST and soil water content (SWC)—explained 44.2% of the variation in the monthly SR rate, whereas biological (i.e., bamboo type) and management (i.e., fertilization) factors had a much smaller impact (less than 0.1% of the variation). The annual mean SR showed a significant positive correlation with soil organic matter (SOM; r = 0.51, P < 0.05), total nitrogen (TN; r = 0.47, P < 0.05), total phosphorus (TP; r = 0.60, P < 0.01), clay content (0.72, P < 0.05) and below-ground biomass (r = 0.60*), which altogether explain 69.0% of the variation in the annual SR. Our results indicate that the fertilization effect was not significant in SR rate for most months among the treatments, but was significant in the annual rate. These results may help to improve policy decisions concerning carbon sequestration and the management of bamboo forests in China.

**Keywords:** red soil; *Phyllostachys glauca*; *Phyllostachys praecox*; partial canonical correspondence analysis

## 1. Introduction

Soil respiration (SR), commonly referred to as the $CO_2$ efflux at the soil surface, is the main channel for carbon transfer from terrestrial ecosystems to the atmosphere [1–3]. The $CO_2$ efflux from soils to the atmosphere has been estimated as 50–70 Pg C year$^{-1}$, accounting for approximately 25% of the total global $CO_2$ emissions; therefore, SR plays a critical role in the global carbon cycle [4–6]. SR greatly contributes to global warming which, in turn, reinforces SR through climate warming effects [7]. Previous studies have speculated that SR may result in the emission of an additional 60 Pg C from the soil into the atmosphere from 1990 to 2050 due to global warming, which is equivalent to a 19% increase in fossil-fuel burning over the same period [2,8]. As an important part of terrestrial ecosystems, forests

are regarded as one of the most important carbon (C) sinks, and carbon emitted from forest soil may greatly affect the atmospheric $CO_2$ concentration [9,10]. Therefore, understanding the carbon cycle in forest ecosystems is important for evaluating the global C budget.

Numerous studies have investigated SR in various forest ecosystems worldwide [10–15], in which high variability in SR has been reported [2,5,11,15]. The degree of SR variation is related to specific site conditions, such as natural factors (including climatic and edaphic conditions), management factors (including fertilization and irrigation), and biological factors (including species type and stand age) [2,11,14,15]. As many studies have suggested, in the presence of no management disturbance, the variation of SR in forest ecosystems can mainly be attributed to natural factors, such as soil temperature (ST) and soil water content (SWC) [2,16]. For example, Sheng et al. [16] conducted an experiment within a representative land-use sequence in the subtropical region of China. The results showed that SR exhibited a distinct seasonal pattern which was dominantly controlled by ST. Similarly, Xu and Qi [2] found that the seasonal variation of SR in an eight-year-old ponderosa pine (*Pinus ponderosa*) plantation could be attributed to ST and SWC. However, with an increase of human influence (including fertilization and irrigation), the effect of natural factors on the variation of SR will decrease while the management effect increases. However, many previous studies have mainly focused on the effects of only one factor (mainly natural factors) on SR variation in forest ecosystems, seldom examining the joint influence of controlling factors on SR variation. Therefore, further studies are required to analyze the quantitative influence and interaction of various factors on the variation of SR for the development of a comprehensive understanding of the factors controlling the patterns of soil $CO_2$ emission in forest ecosystems.

Bamboo forests are among the most important forest types worldwide. The growth patterns of bamboos differ from those of timber, with unique characteristics including fast growth and high production [17–19]. China has rich bamboo resources (>500 species in 39 genera), with a total cultivation area of 6.01 million ha, accounting for 27.3% of the bamboo forest area in the world (data from the Eighth National Forest Resource Inventory Report of China). As important bamboo types, *Phyllostachys praecox* and *Phyllostachys glauca*, which have strong growth and reproduction abilities, as well as high economic value, are widely distributed in South China and may play an important role in the carbon cycle of the forest ecosystem in China. Therefore, we conducted a field experiment in subtropical China for two types of bamboo forest (*Phyllostachys praecox* and *Phyllostachys glauca*), in order to understand the temporal patterns of SR, to determine the major factors for SR variation, and to improve the estimates of the global carbon budget. Accordingly, we hypothesized that soil respiration of bamboo forests was primarily influenced by temperature, but it will be affected by other factors, such as soil moisture, management measures, bamboo species and the objectives of the present study are: (i) to identify the temporal change patterns of SR in two types of bamboo plantations and (ii) to investigate the various factors (including the natural, biological, and management factors) affecting these patterns.

## 2. Materials and Methods

### 2.1. Site Description

The field experiment was conducted at the Ecological Experimental Station of Red Soils, Chinese Academy of Sciences, located in Yujiang County, Jiangxi Province, China (116°5′30′′ E, 28°5′30′′ N). This region features a typical subtropical monsoon climate with an annual mean temperature of 18 °C and annual precipitation of 1785 mm (mainly from March to June). Rainfall and air temperature data from the Farmland Ecosystem of the Yingtan National Field Observation and Research Station, which was adjacent to the experimental plot, for the experimental period are shown in Figure 1. The annual evaporation averaged 1318 mm and the number of frost-free days was 239. The experimental site is 50 m above sea level and is a relatively flat area with slopes of <3°. The red soil of this area can be classified as Ultisols in the Soil Taxonomy System of the U.S.A. and Acrisols and Ferralsols in the FAO legend, and it was derived from quaternary red clay [20].

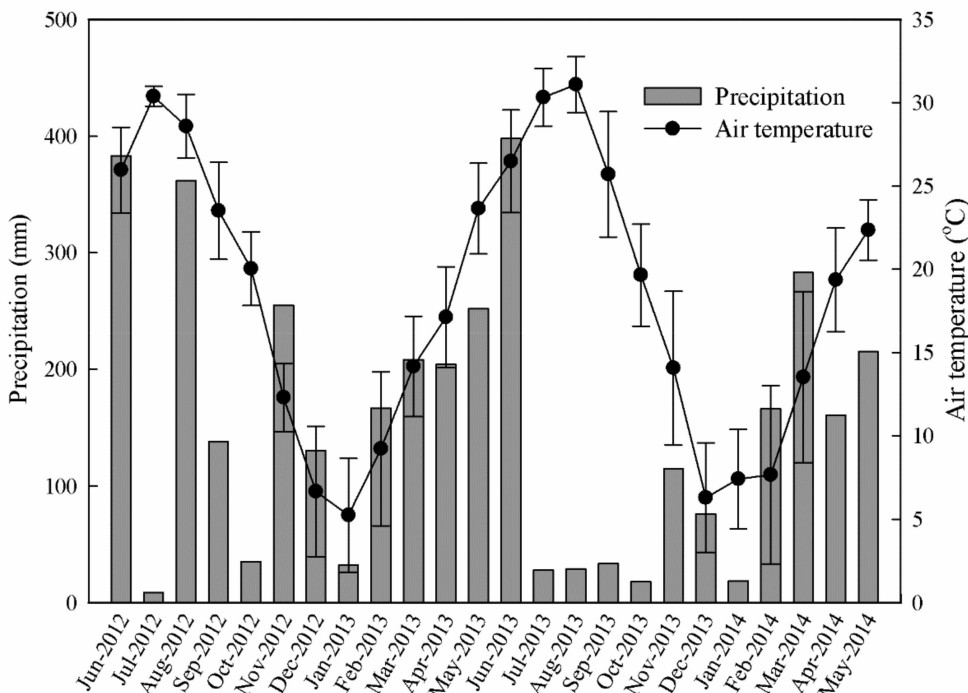

**Figure 1.** Monthly precipitation and air temperature during the experimental period from June 2012 to May 2014 from an automatic weather station (the Ecological Experimental Station of Red Soils, Chinese Academy of Sciences) located at the experimental site. The error bars represent SD of daily air temperature.

*2.2. Experimental Design*

Generally, bamboo farmers apply fertilizers to *Phyllostachys praecox* to attain more bamboo shoots, but not to *Phyllostachys glauca*. According to the cultivation habits of bamboo farmers, we applied fertilization only to *Phyllostachys praecox* in this study. Therefore, there were three treatments for the two types of bamboo forest: (1) *Phyllostachys glauca* with no fertilization (PG), (2) *Phyllostachys praecox* with no fertilization (PP), and (3) *Phyllostachys praecox* with fertilization (PPF). There were nine plots (each of dimensions 5 m × 20 m) for the three treatments; a randomized block design was used to prepare three replicate plots of each of the three treatments in October 2010, where there was a distance of 2 m and a ditch (50 cm in depth) between adjacent plots. *Phyllostachys glauca* (a local bamboo species) and *Phyllostachys praecox* (from Zhejiang Province) were transplanted into the experiment plot in October 2010. The chemical fertilizer used in this experiment was an N–P–K compound fertilizer ($N:P_2O_5:K_2O = 15:15:15$), which was applied to the surface soil (0–3 cm) for PPF treatment at a rate of 800 kg ha$^{-1}$ (a traditional fertilization rate adopted by local bamboo farmers) on October 21, 2012; July 2, 2013; and November 18, 2013. Selected properties of the two types of bamboo and soil prior to measurement are presented in Table 1.

**Table 1.** Characteristics of the bamboo stands in the study sites.

| Stand Type/ Management | Soil Organic Matter (g kg$^{-1}$) | Soil pH | Bulk Density (g cm$^{-3}$) | Sand (%) | Silt (%) | Clay (%) | Soil Texture | Culm Density (culm ha$^{-1}$) | Diameter at Breast Height (cm) | Height (m) |
|---|---|---|---|---|---|---|---|---|---|---|
| PG | 6.95 ± 1.67 | 4.73 ± 0.09 | 1.31 ± 0.11 | 49.68 ± 2.49 | 36.90 ± 1.40 | 13.42 ± 1.26 | Loam | 253667 ± 3118 | 2.05 ± 0.26 | 4.70 ± 0.26 |
| PP | 7.34 ± 1.24 | 4.51 ± 0.04 | 1.37 ± 0.06 | 50.27 ± 3.54 | 34.68 ± 1.79 | 15.05 ± 2.08 | Loam | 9567 ± 503 | 3.21 ± 0.21 | 4.93 ± 0.22 |
| PPF | 7.89 ± 1.14 | 4.50 ± 0.02 | 1.28 ± 0.02 | 45.91 ± 1.39 | 37.29 ± 1.70 | 16.80 ± 0.82 | Loam | 9667 ± 651 | 3.40 ± 0.49 | 5.12 ± 0.33 |

Summary data are followed by mean ± SD (*n* = 3). PG, PP, and PPF represent non-fertilized *Phyllostachys glauca*, non-fertilized *Phyllostachys praecox*, and *Phyllostachys praecox* with fertilization, respectively. Soil organic matter, soil pH, bulk density, and granulometric composition were measured in September 2011. The granulometric composition was classified as sand (2–0.05 mm), silt (0.05–0.002 mm), and clay (<0.002 mm), according to the United States Department of Agriculture standard.

### 2.3. Soil Respiration Measurements

Soil respiration was measured, on a twice-monthly basis, from June 2012 to May 2014 at each plot using a LI-6400 with Li-6000-9 soil chamber and soil temperature probe (LI-COR Inc., Lincoln, NE, USA). The parameters of the target (350–450 μmol mol$^{-1}$ $CO_2$), delta value (5–20 μmol mol$^{-1}$ $CO_2$), chamber insertion depth (0 cm; that is, the chamber was inserted flush with the ground, within the soil collars), and surface area (81.6 cm$^2$) were pre-set before each measurement. Soil collars (polyvinyl chloride collar, 10 cm in diameter and 5 cm in height) were inserted 2 cm into the soil, where they remained for the duration of the experiment. Small plants, litter, insects, and grasses were regularly and carefully removed from each collar. All measurements occurred between 12:00 and 16:00 on each measuring date. The SR rates are expressed as $CO_2$ μmol m$^{-2}$ s$^{-1}$. The ST was measured at 10 cm below the surface by inserting the ST probe near the soil $CO_2$ efflux chamber each time the SR was monitored. The volumetric soil water content (moisture) was measured simultaneously using a Meridian Measurement MPM-160 Moisture Probe Meter (ICT International P/L, Armidale, Australia) with probes (5 cm depth) inserted vertically through the forest floor.

Monthly SR, ST, and SWC were calculated by averaging the two measurements performed within the corresponding month. The cumulative SR for one year was calculated as:

$$M = \frac{\sum (SR_{i+1} + SR_i) \times Day_j \times 24 \times 60 \times 60 \times 12 \times 10^{-8}}{2} \tag{1}$$

where *M* is the cumulative soil respiration (Mg C ha$^{-1}$ year$^{-1}$), *SR* is the soil respiration rate (μmol $CO_2$ m$^{-2}$ s$^{-1}$), *i* is the sample number, *j* is the number of the month in the year, and $Day_j$ is the days of the *j*th month.

### 2.4. Soil Sampling and General Properties

Soil sampling was performed on September 13, 2011; October 19, 2012; and November 18, 2013. Five replicates of the soil cores of a diameter of 5 cm were collected from the surface soil (0–20 cm) of each experiment plot. Volumetric samples were taken from each plot to determine the bulk density. Coarse fragments were removed by hand and the soil samples were air-dried, ground and sieved (2 mm) for further analysis of the soil chemical properties.

The soil granulometric composition was analyzed using the pipette method, the soil pH was measured using the potentiometric method, and the bulk density (BD) was measured using the cutting ring method. The soil organic matter was obtained using the potassium dichromate external heating method. The soil total nitrogen (TN) content was measured using the Kjeldahl digestion method. The soil total phosphorus (TP) content was determined using sulfuric acid–perchloric acid heating digestion and Mo–Sb colorimetric methods. The soil total potassium (TK) content was determined using the inductively coupled plasma method after hydrofluoric acid–perchloric acid digestion [21].

### 2.5. Bamboo Biomass Measurements

Three bamboo samples were obtained from each plot, based on the diameter at breast height (DBH), on July 2, 2013. The fresh weight of three above-ground sections—culms, branches, and foliage—were determined separately. To make it practical to collect the below-ground sections of the bamboo (i.e., living bamboo roots, bamboo rhizomes, and bamboo stump), we assumed that the below-ground sections were evenly distributed under the ground [19]. The quadrat area of a single bamboo plant below ground was the plot area divided by the total number of bamboo plants in the plot (culms). A circle was drawn around the sampled bamboo plant to identify its single quadrat area. Next, the quadrat was excavated to a depth of approximately 60 cm; that is, the depth at which the roots and bamboo rhizomes can reach. In each quadrat, the fresh weight of the living bamboo roots (>5 mm), bamboo rhizomes, and bamboo stump were recorded separately in the field. Samples were taken from each section of the bamboo for dry matter determination. The samples were then oven-dried at 105 °C until they reached a stable weight. Biomass estimation was based on the ratio of the absolute dry weight to the fresh weight. The biomass of a single bamboo plant was calculated by summing the biomasses of all the individual sections of the bamboo plant.

The total culm number of bamboo plants (higher than 2 m) in each plot was counted on August 3, 2013 and July 2, 2013, which was then converted into the culm density (culm ha$^{-1}$). The density of bamboo biomass (kg ha$^{-1}$) for each plot was calculated by the single bamboo biomass multiplied by the culm densities for August 3, 2013 and July 2, 2013, representing the biomass density in each plot in the periods from June 2012 to May 2013 and from June 2013 to May 2014, respectively.

### 2.6. Statistical Analyses

Soil temperature (ST) and soil water content (SWC) have been considered to have a major impact on SR in previous studies [1,2]. Therefore, two types of commonly used models (univariate and bivariate models) were chosen to explore the relationships between monthly SR rate and ST or (ST and SWC). The univariate models included three ST-dependent equations:

$$SR = a \times ST + b \tag{2}$$

$$SR = a \times e^{b \times ST} \tag{3}$$

$$SR = a \times ST^b \tag{4}$$

where *SR* is the soil respiration rate (μmol $CO_2$ m$^{-2}$ s$^{-1}$); *ST* is the soil temperature (°C) at a 10 cm depth; and *a*, *b*, and c are constants fitted to the regression equation.

The bivariate models, based on the ST and SWC, include the two following equations:

$$SR = a \times ST^b \times SWC^c \tag{5}$$

$$SR = a \times e^{b \times ST} \times SWC^c \tag{6}$$

where SWC is the soil water content (m$^3$ m$^{-3}$).

In order to assess the response of the monthly SR rate to temperature change or global climate warming, the apparent temperature sensitivity of the SR ($Q_{10}$), defined as the change of the SR rate with a temperature rise of 10 °C [14,22], was also investigated in this study. According to the parameter b (the temperature reaction coefficient), computed from Equation (3), the $Q_{10}$ value is calculated as:

$$Q_{10} = e^{10b} \tag{7}$$

To quantify the rates of the contribution of various factors to the monthly SR rate and to detect the interaction between the factors, partial canonical correspondence analysis (PCCA) was used in this study. PCCA is an improved version of the canonical correspondence analysis (CCA), which has

been widely used in quantitative ecology. PCCA can perform gradient analysis by ranking techniques and quantify the complex relationships between ecological species and environmental variables [23]. PCCA uses the partitioning variables technique, based on CCA; therefore, PCCA is able to classify the variables into various factors and can calculate the explanatory percentage of the other factors by fixing a factor, thus attaining the absolute contribution rate of various factors on the dependent variable [23]. In addition, PCCA eliminates factors with collinearity, according to variance inflation factors (VIF), to ensure the mutual independence of the factors involved in the calculation. Therefore, PCCA is able to quantify the contributions of various factors and the interactions between them to the dependent variable. To quantify the influences of the various environmental variables on the monthly SR rate in the studied bamboo forests, PCCA was performed based on a "species matrix" in this study. The species matrix was constructed based on the mean SR rate and the coefficient of variation calculated from the four observed SR within every two months. The environmental variables included ST, SWC, fertilization, and bamboo type. The calculation result showed that the VIF values of the four environmental variables were all smaller than 20; therefore, all four variables were included during the analysis (Table 2). In this study, PCCA analysis was conducted using the "Vegan" package in R language 2.7.1 (version, website).

**Table 2.** Influential variables (variance inflation factors < 20) and their correlations with the monthly soil respiration rate.

| Variable | Variance Inflation Factors (VIF) | *P* Value |
|---|---|---|
| Soil temperature | 1.35 | 0.001** |
| Soil water content | 1.35 | 0.551 |
| Bamboo type | 1.41 | 0.672 |
| Fertilization | 1.39 | 0.909 |

** represents extremely significant, $P < 0.01$.

Statistical analyses (including correlation and regression) were completed using the SPSS 17.0 (SPSS Inc., Chicago, IL, USA) software. The coefficient of variation (CV) was used to represent the temporal variation of the SR measured at different times. One-way Analysis of Variance (ANOVA) with Fisher's LSD tests was used to test the difference among the three treatments in SR, ST, and SWC, as well as the soil properties. The significance was determined at the $P = 0.05$ level. The figures were plotted in using the Sigma plot 10.0 (Systat Software Inc., Chicago, IL., USA) and Matlab 7.0 (MathWorks Inc., Natick, MA., USA) software.

## 3. Results

### 3.1. Variation of the Monthly SR Rate and Annual SR

From November 2012 to November 2013, the monthly SR rates were lowest during winter, gradually increasing from late spring and peaking in summer, with SR rates reaching 2.62, 3.25, and 3.87 μmol $CO_2$ m$^{-2}$ s$^{-1}$ in June for the PG, PP, and PPF plots, respectively. However, an abrupt 'drop' in the SR occurred after June in 2013, when the soil experienced low water content (SWC <0.15 m$^3$ m$^{-3}$ soil) resulting from continuous high temperatures and scarce rainfall (Figures 1 and 2). This SR pattern in the summer (June to September) of 2013 was different from that in the summer of 2012 when such a soil water limitation did not occur (Figure 2a,b). In addition, the monthly dynamic pattern was different for SR and ST in PPF, especially during the winter in 2013 (after the third fertilization). The ST showed a similar monthly pattern to that of the SR, especially for the PG and PP treatments, reaching a maximum value of 27.8, 29.8, and 30.0 °C for the PG, PP, and PPF plots, respectively, in August, when the SWC reduced to its minimum values of 0.08, 0.09, and 0.07 m$^3$ m$^{-3}$, respectively (Figure 2b). The SWC did not exhibit any large fluctuations from June 2012 to April 2013 and had an opposite trend to soil temperature from May 2013 to May 2014 (Figure 2b). Although PPF had a higher SR rate than PP and PG in most months (especially for the months between November

2013 and January 2014, when the third fertilization was applied), the SR rate, ST, and SWC were not significantly different among the treatments during most of the study period. The CV of the monthly soil respiration rate for PG, PP, and PPF were 0.30, 0.33, and 0.31, respectively.

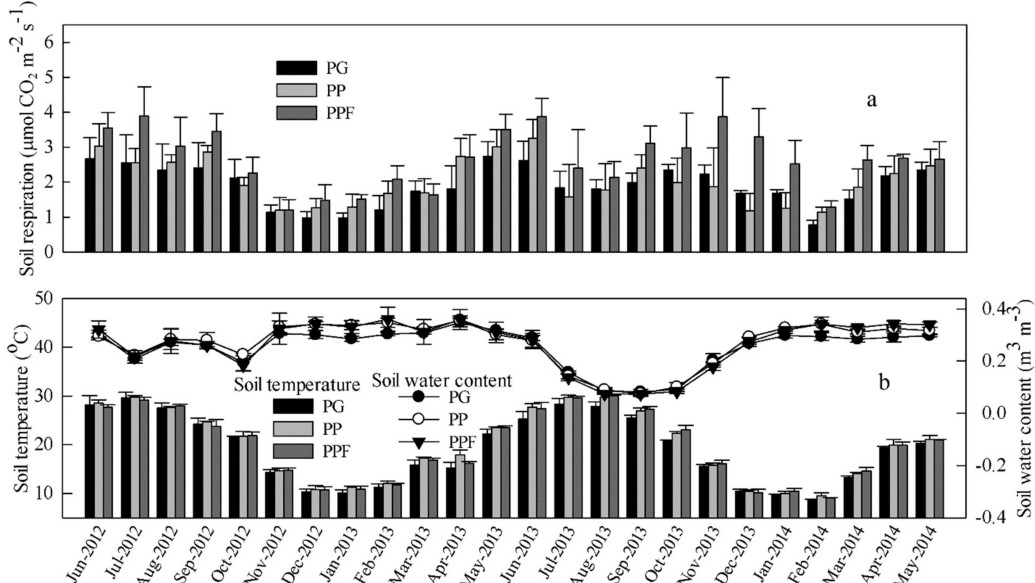

**Figure 2.** The monthly pattern of the soil respiration rate (**a**) and the soil temperature (at 10 cm) and the water content (at 5 cm) (**b**) for the three treatments during the experimental period (June 2012 to May 2014). PG, PP, and PPF represent non-fertilized *Phyllostachys glauca*, non-fertilized *Phyllostachys praecox*, and *Phyllostachys praecox* with fertilization, respectively. The vertical bar represents the standard deviation (mean ± SD, $n$ = 3).

The cumulative soil respiration over the two year period was 14.44, 15.40, and 20.14 Mg C ha$^{-1}$ for the PG, PP, and PPF plots, respectively, corresponding to mean annual carbon flux rates of 7.22, 7.70, and 10.70 Mg C ha$^{-1}$ year$^{-1}$, respectively (Figure 3). The annual SR in PPF was significantly higher than that in PG and PP, whereas there was a significant difference in the annual SR observed between the PG and the PP.

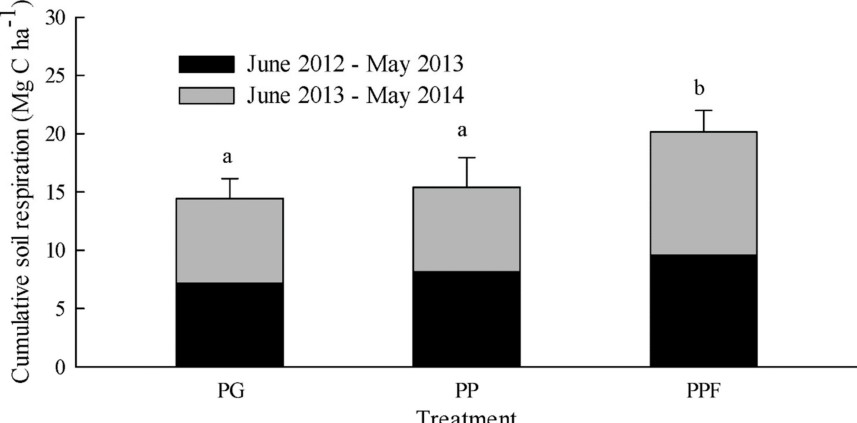

**Figure 3.** The cumulative soil respiration for the three treatments during the two experimental years (June 2012 to May 2013 and June 2013 to May 2014) in subtropical China. PG, PP, and PPF represent non-fertilized *Phyllostachys glauca*, non-fertilized *Phyllostachys praecox*, and *Phyllostachys praecox* with fertilization, respectively. The error bars represent SD of accumulative SR over the two years (June 2012–May 2014) for the three replicates of each treatment and the same character means there is no difference between the treatments.

### 3.2. Statistical Results of the Monthly SR Rate Versus the ST and the SWC

A significantly positive correlation (linear) between the monthly SR rate and the ST was found for all three treatments at a monthly scale, whereas no significant correlation was observed between the monthly SR rate and the SWC (Table 3). The correlation coefficients between the monthly SR rate and the ST for the three treatments ranged from 0.4–0.6.

**Table 3.** Results of the Pearson correlation analysis between the soil respiration rate and the soil temperature or the soil water content at the monthly scale for the bamboo forest.

|  | Treatment | Soil Temperature | Soil Water Content |
|---|---|---|---|
| Soil respiration | PG | 0.595 ** | −0.132 |
|  | PP | 0.555 ** | −0.066 |
|  | PPF | 0.396 ** | −0.149 |

PG, PP, and PPF represent non-fertilized *Phyllostachys glauca*, non-fertilized *Phyllostachys praecox*, and *Phyllostachys praecox* with fertilization, respectively. The number of observations for monthly soil respiration was 144. ** Significant at the P = 0.01 level.

The result of the univariate regressions showed that the monthly variation of the SR rate could be explained by the ST, with $R^2$ values ranging from 0.148–0.373 in the three treatments (Table 4). Compared with the univariate regressions, the bivariate models (integrating the monthly SR rate against the ST and the SWC) yielded higher $R^2$ values by various margins for the three treatments. The best-fit bivariate regression for the monthly variation of the SR rate in the three treatments was based on Equation (4), with $R^2$ values of 0.412, 0.435, and 0.196 for PG, PP, and PPF, respectively (Table 4). Compared with PG and PP, the PPF treatment had smaller $R^2$ values for the selected models for the monthly scale. The $R^2$ values of the best-fit bivariate models in PG, PP, and PPF increased by 0.10, 0.14, and 0.06, respectively, when excluding the extreme weather data from July to October in 2013; that is, when the soil had a very low water content due to continuous high air temperatures and scarce rainfall (see Figure 4).

Calculation of $Q_{10}$ at the monthly scale based on the ST showed that the $Q_{10}$ values were 1.35, 1.34, and 1.27 in PG, PP, and PPF, respectively. When regrouping the monthly data (including SR rate and ST) based on the SWC, the $Q_{10}$ values were 0.85, 0.82, and 1.17 in PG, PP, and PPF with SWC < 0.19, respectively; and 1.49, 1.52, and 1.40 in PG, PP, and PPF with SWC ≥ 0.19, respectively.

**Table 4.** Univariate and bivariate models of the soil respiration rate against the soil temperature (at 10 cm) and the soil water content (at 5 cm) at the monthly scale.

| Model Type | Equation | $R^2$ | | |
|---|---|---|---|---|
|  |  | PG | PP | PPF |
| Univariate model | $SR = a \times ST + b$ | 0.354 | 0.308 | 0.157 |
|  | $SR = a \times e^{b \times ST}$ | 0.320 | 0.275 | 0.148 |
|  | $SR = a \times ST^b$ | 0.373 | 0.326 | 0.167 |
| Bivariate model | $SR = a \times ST^b \times SWC^c$ | 0.412 | 0.435 | 0.196 |
|  | $SR = R_0 \times e^{a \times ST} \times SWC^b$ | 0.353 | 0.377 | 0.179 |

PG, PP, and PPF represent non-fertilized *Phyllostachys glauca*, non-fertilized *Phyllostachys praecox*, and *Phyllostachys praecox* with fertilization, respectively. The number of observations for the monthly soil respiration rate was 144. $R_0$, a, b, and c are the parameters for the equations. $R^2$ is the coefficient of determination for the model.

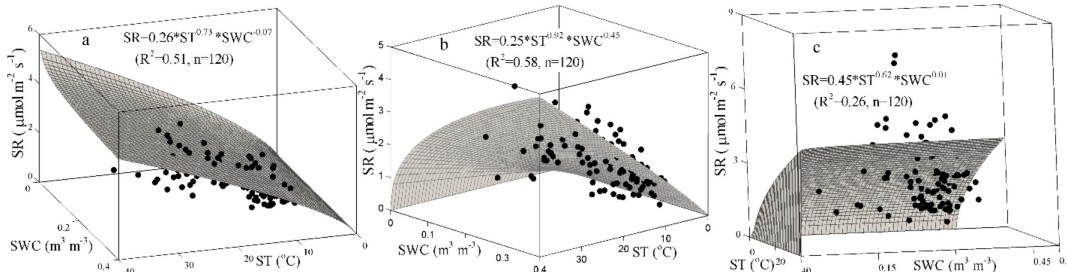

**Figure 4.** The best-fit bivariate models based on the soil temperature (ST) and the soil water content (SWC) for the monthly soil respiration (SR) rate for the three treatments in subtropical China: non-fertilized *Phyllostachys glauca* (**a**), non-fertilized *Phyllostachys praecox* (**b**), and *Phyllostachys praecox* with fertilization (**c**).

### 3.3. PCCA Results of the Monthly SR Rate with Various Factors

The results show that the linear correlation coefficients of these four variables (ST, SCW, bamboo type, and fertilization) were small and, thus, could be used as independent variables for PCCA. The result of the factor variance analysis (Permutation Test, Permutations = 200) demonstrated that the ST had an extremely significant influence on the variation of monthly SR rate, whereas the other variables (SWC, bamboo type, and fertilization) showed no significant influence on SR rate (Table 2). In order to quantify the effects of various factors on the monthly SR rate, the variables first needed to be classified. ST and SWC represented climate effects (natural factor) on the SR, while bamboo type represented the biotic factor and fertilization represented the management factor. Therefore, in this study, the four variables were classified into three factors: a natural factor (ST and SWC), a management factor (fertilization), and a biotic factor (bamboo type). Individual effects and interactions between the factors on the monthly SR rate were calculated based on these three factors.

The results indicated that the natural factor alone could explain 44.22% of the variation in the monthly SR rate in bamboo forests, while the biotic and management factors could explain only a small part of the variation of the monthly SR rate, with values of 0.03% and 0.07%, respectively (Figure 5). The interaction between the biotic factor and the management factor on the variation of monthly SR rate was 0.00%, and the interactions between the natural factor and the biotic and management factors were 0.31% and 0.07%, respectively (Figure 5). Hence, other unknown factors explained 55.40% of the variation of the monthly SR rate (Figure 5). Therefore, among the three factors, the natural factor (including the ST and the SWC) had the largest influence on the variation of the monthly SR rate in the bamboo forests.

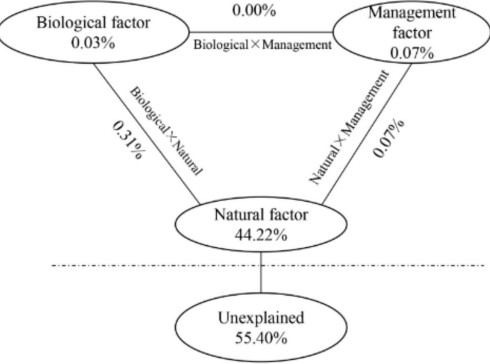

**Figure 5.** Contributions of biological, management, and natural factors to the monthly soil respiration rate in the bamboo forests in Yingtan, China, as revealed by partial canonical correspondence analysis (PCCA).

### 3.4. Soil Properties and Bamboo Biomass on the Annual SR

The soil properties of the three treatments measured on October 19, 2012 and November 18, 2013, representing the mean soil properties of the first year (June 2012 to May 2013) and the second year (June 2013 to May 2014), respectively, are summarized in Table 5. During the second experimental year, significant differences in BD, pH, and clay among the three treatments were observed, whereas no significant differences were observed in other properties among the three treatments.

**Table 5.** Summary of the soil properties for every treatment measured on October 19, 2012 and November 18, 2013.

| Time | Treatment | BD (g cm$^{-3}$) | Soil pH | SOM (g kg$^{-3}$) | TN (g kg$^{-3}$) | TP (g kg$^{-3}$) | TK (g kg$^{-3}$) | Sand (%) | Silt (%) | Clay (%) |
|---|---|---|---|---|---|---|---|---|---|---|
| 20121019 | PG | 1.28 ± 0.11A | 4.67 ± 0.02A | 7.08 ± 1.02A | 0.53 ± 0.05A | 0.57 ± 0.10A | 20.50 ± 0.75A | 46.40 ± 2.63A | 39.45 ± 2.65A | 14.15 ± 0.70A |
| | PP | 1.26 ± 0.15A | 4.69 ± 0.08A | 8.80 ± 1.76A | 0.51 ± 0.07A | 0.59 ± 0.11A | 20.22 ± 1.46A | 43.23 ± 2.42A | 41.40 ± 3.50A | 15.37 ± 2.30A |
| | PPF | 1.28 ± 0.02A | 4.60 ± 0.03A | 8.56 ± 1.69A | 0.58 ± 0.04A | 0.58 ± 0.07A | 19.85 ± 0.86A | 43.65 ± 3.86A | 39.93 ± 1.68A | 16.41 ± 2.31A |
| 20131118 | PG | 1.57 ± 0.14a | 4.93 ± 0.06a | 6.53 ± 2.94a | 0.41 ± 0.06a | 0.52 ± 0.10a | 18.09 ± 0.68a | 42.59 ± 4.09a | 43.61 ± 3.37a | 13.80 ± 0.73a |
| | PP | 1.37 ± 0.06b | 4.80 ± 0.08b | 7.11 ± 2.73a | 0.51 ± 0.09a | 0.57 ± 0.12a | 18.44 ± 1.03a | 41.25 ± 4.77a | 42.91 ± 5.49a | 15.84 ± 1.18ab |
| | PPF | 1.52 ± 0.06ab | 4.76 ± 0.05b | 7.63 ± 2.97a | 0.53 ± 0.15a | 0.67 ± 0.14a | 18.75 ± 1.63a | 40.72 ± 2.42a | 41.77 ± 1.50a | 17.51 ± 1.69b |

PG, PP, and PPF represent non-fertilized *Phyllostachys glauca*, non-fertilized *Phyllostachys praecox*, and *Phyllostachys praecox* with fertilization, respectively. Summary data are followed by mean ± SD (*n* = 3) for all of the soil properties. BD, SOM, TN, TP, and TP represent bulk density, soil organic matter, total nitrogen, total phosphorus, and total potassium, respectively. The granulometric composition was classified as sand (2–0.05 mm), silt (0.05–0.002 mm), and clay (<0.002 mm), according to the United States Department of Agriculture standard. Significant at the *P* = 0.05 level.

The culm density of the bamboo plantation during the two experimental years was significantly greater in PG (25367–28267 culm ha$^{-1}$) than that in PP (9967–12367 culm ha$^{-1}$) and PPF (9933–13233 culm ha$^{-1}$), whereas the single culm biomass was significantly smaller in PG (0.49 ± 0.07 kg culm$^{-1}$) than PP (0.87 ± 0.03 kg culm$^{-1}$) and PPF (0.97 ± 0.05 kg culm$^{-1}$). An increase in the total biomass density of the bamboo (both above-ground and below-ground) was observed from the first year (June 2012 to May 2013) to the second year (June 2013 to May 2014), ranging from 12.40 ± 0.65 to 13.86 ± 0.98 Mg ha$^{-1}$ in PG, from 8.71 ± 0.95 to 10.81 ± 1.23 Mg ha$^{-1}$ in PP, and from 9.64 ± 1.22 to 12.82 ± 0.84 Mg ha$^{-1}$ in PPF. Although PG had a higher total biomass density of bamboo than PP and PPF, the below-ground biomass of PPF was significantly greater than that of PG for the first year and higher than that of PP and PG for the second year (Figure 6).

Pearson correlation analysis showed that there was a significant correlation between the annual SR and the below-ground biomass (0.60*), SOM (r = 0.51*), TN (r = 0.47*), TP (r = 0.60*), and clay (r = 0.72*) in the three treatments, while no significant correlation was observed between the annual SR and the other soil properties. Regression analysis demonstrated that the below-ground biomass, SOM, TN, TP, and clay collectively explained most of the variability in the annual SR, where the regression is given by Equation (8):

$$Annual\ SR = 1.3 \times BB + 0.3 \times SOM - 0.9 \times TN + 1.6 \times TP + 0.25 \times Clay - 2.4\ (R^2 = 0.69, n = 118, P < 0.01) \tag{8}$$

where SR, BB, SOM, TN, and TP are soil respiration (µmol $CO_2$ m$^{-2}$ s$^{-1}$), below-ground biomass (kg ha$^{-1}$), soil organic matter (g kg$^{-1}$), total nitrogen (g kg$^{-1}$), and total phosphorus (g kg$^{-1}$), respectively.

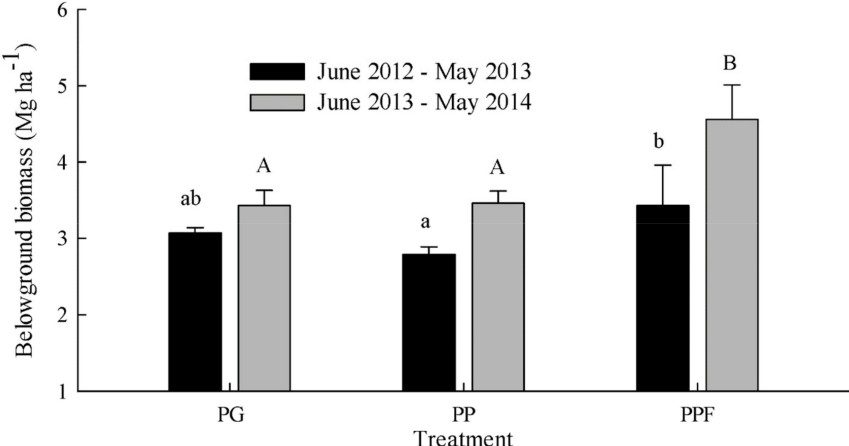

**Figure 6.** Below-ground biomass in the bamboo plantation during the two experimental years (June 2012 to May 2013 and June 2013 to May 2014) for the three treatments: non-fertilized *Phyllostachys glauca* (PG), non-fertilized *Phyllostachys praecox* (PP), and *Phyllostachys praecox* with fertilization (PPF). The error bars represent SD of annual belowground biomass for each treatment. Different lowercase letters and uppercase letters indicate a significant difference for treatments in the year of June 2012 to May 2013 and in the year of June 2013 to May 2014, respectively.

## 4. Discussion

### 4.1. Effect of the ST and the SWC on the Monthly SR Rate

The rates of SR (0.38–8.53 $\mu$mol $CO_2$ m$^{-2}$ s$^{-1}$) presented herein fell within the range of SR rates which have been reported by numerous studies for forests worldwide [13,14,24,25]. The variation pattern of the monthly SR rate was distinct in all treatments in the present study, which was consistent with the observations of other studies on forest SR rates in subtropical areas with obvious seasonal climatic characteristics [10,14,16,24]. In general, there was a strong correspondence between the SR rate and the ST, especially for the PG and PP treatments at a monthly time scale throughout the present study, which was consistent with numerous previous studies [1,11,14,24]. Univariate regression analysis showed that the ST alone could explain 0.33 and 0.37 of the variation of the monthly SR rate in PG and PP, respectively (Table 4). This result was in accordance with the results reported in other studies, which suggested that the ST affects the pattern of the monthly SR rate mainly by regulating the plant and microbial activities related to soil carbon cycling [13,16]. However, there was a discordant trend between the SR rate and the ST during the summer in 2013, in which an abrupt 'drop' in the SR rate was observed after June of 2013, with a high ST and a low SWC (<0.15). The phenomenon that the SWC weakened the effect of the ST on the variation of SR rate during the mid-summer drought confirmed similar observations made by Xu and Qi [2], who reported that the limitations on the SR rate in drought may result from reduced microbial activity due to soil water deficits. Furthermore, a minimum SWC may be required for microbial activity in the decomposition processes. Therefore, as was found in our study, the SR rate can be better explained using bivariate regression which includes SWC at the monthly scale. Furthermore, some studies have suggested that the relationship between soil respiration and soil temperature varies according to moisture thresholds [26,27], thus when the data from July to October 2013 were excluded, due to the limited soil water, the best-fit bivariate models further enhanced the explanatory value of the variation of the monthly SR rate (Figure 4). The apparent temperature sensitivity ($Q_{10}$) of the soil respiration has been shown to be widely variable among different ecosystems [11,14,16]. The $Q_{10}$ values (0.82–2.86) presented in this paper fell within the range of the SR rates previously reported by numerous studies for forests worldwide [11,13,16]. Our results show that PPF had a lower $Q_{10}$ value, compared with PP and PG, suggesting that the application of fertilizer decreased the $Q_{10}$ values in the bamboo plantation of the present study. Similar results have been reported by other studies [13,28]. Mo et al. [28] found that the $Q_{10}$ values in high-N situations

decreased after three years of N fertilizer application in a mature tropical forest, and Tu et al. [13] found that N addition reduced the $Q_{10}$ values in the total SR, as well as the SR derived from plant root soil and root-free soil. Variation of the $Q_{10}$ value may reflect a change of metabolism of the plant roots and soil microbes under N-rich situations. Higher $Q_{10}$ in the bamboo soils with no fertilization indicated that the bamboo stands without fertilization were potentially more sensitive to global warming and, thus, would offset the beneficial effects of increased soil C storage resulting from the decreased SR.

In this study, we found that the $Q_{10}$ value was lower for the data with low SWC (<0.19), than for the data with high SWC (≥0.19), which was similar to the results reported by previous studies [29,30]. These previous studies suggested that a higher $Q_{10}$ value may be attributed to elevated root and microbial respiration. In contrast, the lower $Q_{10}$ value observed during drought periods may have resulted from the reduction in labile substrates for respiration under dry conditions; similarly, desiccation stress of respiring plant tissues or micro-organisms due to drought may also lead to a lower $Q_{10}$. Therefore, we can conclude that the SWC affects the temperature sensitivity; this factor should be taken into account when using the value of $Q_{10}$ to evaluate the SR change in response to global warming, especially at the landscape scale [14,31].

### 4.2. Quantitative Influences of Various Factors on the Monthly SR Rate

The quantitative influences of the individual factors, as well as the interactions of the various factors, on the variation of the monthly SR rate can be determined through PCCA. As shown in the results, the natural factor (including ST and SWC) had a larger influence on the variation of the monthly SR rate in the bamboo forest than the biotic and management factors, which was consistent with the result that the correlation analysis of ST and SWC could explain most of the variation of the SR rate at the monthly scale. Therefore, the main controlling factor that influenced the variation of the monthly SR rate was the natural factor (ST and SWC), in agreement with the results reported in other studies [13,16]. The PCCA results showed that the biotic factor (bamboo type) played a small role in the variation of the monthly SR rate in the bamboo forest, which was consistent with the significance analysis presented above, showing that there was no significant difference in the monthly SR rate between the non-fertilized *Phyllostachys glauca* and the non-fertilized *Phyllostachys praecox*. As shown above, although PPF exhibited a significantly higher annual SR than PP and PG, suggesting that fertilization increased annual SR, the PCCA results demonstrated that the management factor (fertilization) had only a small influence on the variation of the monthly SR. The reason for this result may be that the fertilization effect is not apparent in the SR rate at the monthly scale but might be reflected by the accumulated SR (such as that at the annual scale). The PCCA was performed based on data at a monthly scale (two months); therefore, the results showed that the management factor (fertilization) displayed a small contribution on the variation of the monthly SR. Similar results have also been reported in other studies, suggesting that nitrogen fertilization increased the cumulative soil $CO_2$ efflux [32,33].

However, there was still a considerable fraction (55.40%) of the variation of the monthly SR explained by other unknown factors, which may be attributed to the complex conditions of the bamboo forest in the field.

### 4.3. Effect of Fertilization, Bamboo Biomass, and Soil Properties on the Annual SR

The annual SR ranged from 7.22 to 10.70 Mg C ha$^{-1}$ year$^{-1}$ in the three treatments in the experimental bamboo plantation in subtropical China. On average, the soils in our bamboo plantation released approximately 8.54 ± 1.88 Mg C ha$^{-1}$ year$^{-1}$ into the atmosphere; this value was within the range of annual SR from forest soil, as reported by previous studies [15,16,34].

As discussed above, the effect of fertilization on the SR rate was not significant for most months but could be reflected by the accumulated SR. Among the three treatments, PPF exhibited a significantly higher annual SR than PP and PG, which may be attributed to the application of fertilizers. Our results were similar to the results observed in many other studies, which suggests that fertilizer application

altered the emission of $CO_2$ by regulating different sources of $CO_2$ efflux from the soil, such as plant root respiration, microbial respiration, and soil processes related to carbon and nitrogen cycling [13,35,36]. The magnitude of the total soil carbon emission has been determined by the response of different SR components to fertilization [13]. Furthermore, it has been estimated that root respiration accounts for a large part of the total SR, ranging from 33–62% depending on the condition and type of forest ecosystems [37–39]. In our study, the below-ground biomass, including the roots, had a significantly positive correlation with the SR, which alone could explain 0.36% of the variation of SR ($P < 0.01$). Therefore, it was speculated that the larger below-ground biomass in PPF, as compared to that of PG and PP, was stimulated by fertilization; which, in turn, led to higher SR. Tu et al. [13] also reported that increased available N in the soil from fertilization may promote the growth of fine roots, especially in N-limited forest ecosystems, which leads to the strengthening of plant metabolism and respiration. Meanwhile, as reported by Shao et al. [33], elevated root production can promote substrate availability and microbial activity, leading to increased SR. In addition, previous studies have suggested that the annual SR was affected greatly by the soil organic carbon content [15]. Samuelson et al. [40] and Shao et al. [33] reported that the application of fertilizer enhances soil C by simulating plant growth, which increases the addition of organic carbon to the soil by litter-fall, root turnover, and root exudation, leading to a higher SR. However, there was some uncertainty regarding the estimation of the below-ground biomass in the bamboo plantation in the present study. To ensure continuous measurement of the SR during the experimental period, we did not sample the below-ground biomass under the soil collar and used only the mean estimated below-ground biomass of three culms of the total sampled bamboo to represent the below-ground biomass for the corresponding plot. However, as the bamboo sampled in each plot was adjacent to the location where the soil collar was fixed, the approximation of the below-ground biomass estimated by these samples should, to some extent, represent realistic values.

In addition, a disparity in the monthly change between SR and ST in PPF was observed in the present study, which indicated that fertilization weakened the ST-dependent relationship with SR. Previous studies have also reported that the ST or SWC dependence of SR was altered by fertilizer application [31,41], the sensitivity of soil respiration to temperature significantly increased with C input from fertilizer [42].

## 5. Conclusions

Monthly soil respiration rate showed a similar pattern in all the treatments used in our study, driven largely by soil temperature at a depth of 10 cm, especially for the PG and PP treatments. The disparity in the monthly dynamic pattern between SR and ST in PPF implied that fertilization may greatly affect the ST dependency of SR. Further, SWC was found to play a weaker role in explaining the variation of SR in the experimental bamboo plantation but may weaken the effect of ST on SR variation during mid-summer drought conditions. The best-fit bivariate (integrating SR against ST and SWC) better explained the variation of the SR rate at a monthly scale, as compared to the best-fit univariate (based only on ST). The $Q_{10}$ value of SR was higher with SWC $\geq$ 0.19 than that with SWC $<$ 0.19, suggesting that SWC affected not only SR but also its sensitivity to ST, so in the bamboo forests management we could reduce the SR by irrigation like additional pulse intra-soil watering. PCCA results showed that the natural factor (including ST and SWC) had a larger influence on the variation of the monthly SR rate in the experimental bamboo forest than the biotic (bamboo type) and management (fertilization) factors. PPF showed a significantly higher annual mean SR than PG and PP, suggesting that fertilization could effectively enhance annual SR. The annual mean SR showed significant positive correlations with SOM, TN, TP, clay, and below-ground biomass, and the five variables together could explain a considerable part of the variation of the annual SR. These findings contribute to a more comprehensive understanding of the factors controlling the patterns of soil $CO_2$ emission from bamboo forests in different temporal scales in subtropical China.

**Author Contributions:** "conceptualization, H.Z. and S.Z.; methodology, S.Z. and H.Z.; software, Z.Q.; validation, S.Z. and H.Z.; formal analysis, S.Z. and Z.Q.; investigation, S.Z. and H.Z.; resources, S.Z.; data curation, Z.Q.; writing—original draft preparation, H.Z.; writing—review and editing, H.Z., S.Z. and Z.Q.; visualization, Z.Q. and H.Z.; supervision, S.Z.; project administration, S.Z.; funding acquisition, S.Z. All authors have read and agreed to the published version of the manuscript.

**Funding:** This work was supported by the National Natural Science Foundation of China (31901298 and 41671296) and Natural Science Foundation of Fujian Province (2017J05042). We thank the Farmland Ecosystem of the Yingtan National Field Observation and Research Station for providing the rainfall and air temperature data. We also appreciate the constructive suggestions on the management of bamboo from Professor Renyi Gui of the Zhejiang Agriculture and Forestry University.

**Acknowledgments:** We are grateful for experimental support from Yuhe Zhang and Qiantang Zhang.

**Conflicts of Interest:** The authors declare no conflict of interest.

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
