# Peer review of "Effects of Soil Temperature, Water Content, Species, and Fertilization on Soil Respiration in Bamboo Forest in Subtropical China"

_forests, doi:10.3390/f11010099_

Round 1
Reviewer 1 Report
Highly qualified contribution, suits the journal’s aim and scope well.
Some propositions
Line 96
According to the cultivation habits of bamboo farmers… Therefore, there were three treatments…
It seems that experiment has repeated the standard scheme of fertilization of bamboo plantation. Why you didn’t use the farmer’s plantation for your research? There would be no need to work hard in the plantation yourself. Randomized block design of the experimental plot isn’t sufficient reason for this. It would be better for you to use the regular block design, and spread the individual studied blocks throughout the plantation according the farmer's fertilization scheme, not trying to place the blocks of your experimental scheme necessarily near one another.
Line 139
…the soil was ground by passing…
May be a word “sieved” should be used in this sentence?
The first step – soil was ground, the second step – soil was sieved.
Lines 144-148
The soil total nitrogen (TN) content … The soil total phosphorus (TP) content … The soil total potassium (TK) content…
It is not enough from the plant nutrition view, the forms of nutrients are important too.
Lines 157-158
60 cm; that is, the depth at which the roots and bamboo rhizomes can reach.
It is very important observation. It would be interesting to know the origin of the rhizosphere depth which is linked to the properties of soil, at least, its ability to provide the root penetration.
Lines 171-172
two types of commonly used models (univariate and bivariate models) were chosen
Why don’t you give your preliminary assessment which model is better?
Lines 269-270
the soil had a very low water content due to continuous high air temperature and scarce rainfall
According the climatic data, “annual precipitation of 1785 mm (line 83), annual evaporation averaged 1318 mm (line 86), your experimental plot can be characterized as humid. But according the lines 269-270 you state the need for addition soil moisture. In Introduction section you discuss the irrigation. It would be useful your assessment of the prospects of bamboo irrigation, and on our opinion your propositions in this way will become more weighable if you will use our new water use paradigm: https://plan.core-apps.com/acs_sd2019/abstract/6dc4fdd9-0f1c-41b3-a54e-ac4e18736726
Lines 301-303
The interaction between the biotic factor and the artificial factor on the variation of monthly SR rate was 0.00%, and the interactions between the natural factor and the biotic and artificial factors were
0.31% and 0.07%,…
Statistical procedures authors used are attractive, and useful for experiment which conditions were strictly determined. But for the uncertain processes in natural conditions this methodology gives results which are at least strange. For some factors the procedure provides high accuracy calculation, but helplessly states the unexpected variation value of 55.40 %. If anyone will use the last result directly, it means that experiment was failed. Of course we do not think this way, but the fact mentioned, as well as a low correlation of studied factors needs addition explanation. The author’s expert opinion is needed to assess the used statistical procedures prospects, and possibilities to reveal the unknown factors of variation. It will help to make the future results more certain.
May be, the conclusion ought to contain the author’s proposition how to reduce the SR via additional pulse intra-soil watering.
One more important thing is a carbon dioxide use by plants as the way of carbon dioxide reversible sequestration, which helps to increase the volume of biological matter and the climate biological buffering. It would be useful to mention not only a standard assessment of carbon as a green house gas, but the fact that its green house capability is lower than danger for climate from the water vapor – most dangerous green house gas, which careful management, together with the biological matter management in focus of food, raw materials, bio-fuel production and climate stabilization are the real but very complicated tasks.
Author Response
Dear Sir,
Thank you very much for handling the review of our manuscript (forests-684464) entitled “Effects of soil temperature, water content, species, and fertilization on soil respiration in bamboo forest in subtropical China”. We appreciate the insightful comments and suggestions of anonymous reviewers as well as the subject editor, and have carefully considered each point brought up.
Based on the comments we received, careful modifications have been made to the original manuscript. Below you will find our point-by-point responses to the reviewer’s comments/questions.
Please note that: Reviewer’s comments/questions are in black. Our responses are in blue.
Sincerely yours,
Shunyao Zhuang, Ph.D.
1.Line 96
According to the cultivation habits of bamboo farmers… Therefore, there were three treatments…
It seems that experiment has repeated the standard scheme of fertilization of bamboo plantation. Why you didn’t use the farmer’s plantation for your research? There would be no need to work hard in the plantation yourself. Randomized block design of the experimental plot isn’t sufficient reason for this. It would be better for you to use the regular block design, and spread the individual studied blocks throughout the plantation according the farmer's fertilization scheme, not trying to place the blocks of your experimental scheme necessarily near one another.
Reply: Yes, we agree that the result would be more convincing if we use the farmer ’s plantation for our research directly. However, in order to make the results with different treatments more comparable, we need to make sure that the initial conditions of plots from different treatments are relatively similar as much as possible. Therefore, we keep all the plots with different treatments within a small area where the soil spatial heterogeneity from different plots are smaller than that of the plots we selected from the farmers’ plantation. As for the plot setting, we used a randomized block design to prepare three replicate plots of each of the three treatments, where there was a distance of 2 m and a ditch (50 cm in depth) between adjacent plots. Therefore, our plot setting scheme can minimize the effect of spatial heterogeneity of soil and avoid the interaction between plots as much as possible.
2.Line 139
…the soil was ground by passing…
May be a word “sieved” should be used in this sentence?
The first step – soil was ground, the second step – soil was sieved.
Reply: Thank you for your suggestion and we have used the word “sieved” in the manuscript of modified version.
3.Lines 144-148
The soil total nitrogen (TN) content … The soil total phosphorus (TP) content … The soil total potassium (TK) content…
It is not enough from the plant nutrition view, the forms of nutrients are important too.
Reply: Yes, we agree with your valuable opinion, we also measured indicators such as available phosphorus, available potassium, and the available phosphorus, available potassium showed good correlation with total phosphorus and total potassium, respectively. Thus, the total nutrition was chosen to conduct the correlation analysis with soil respiration to decrease workloard.
4.Lines 157-158
60 cm; that is, the depth at which the roots and bamboo rhizomes can reach.
It is very important observation. It would be interesting to know the origin of the rhizosphere depth which is linked to the properties of soil, at least, its ability to provide the root penetration.
Reply: Bamboo rhizomes were distributed in the range of 0-60 cm but mainly distributed in 0-20 cm, so we collected soil samples from the surface soil (0–20 cm), and we appreciate your valuable suggestion, and we will do further work such as investigate origin of the rhizosphere depth in the future.
5.Lines 171-172
two types of commonly used models (univariate and bivariate models) were chosen
Why don’t you give your preliminary assessment which model is better?
Reply: Thank you for your suggestion and we have added preliminary assessment in the manuscript of revised version. As many results show that soil temperature is one of the most important environmental variables for soil respiration, thereby the temperature-dependence model was most frequently used for estimating soil CO2 fluxes in many studies. However, some other studies also indicate that, in addition to the temperature response, soil respiration would be significantly limited both by low and high soil water contents (SWC) (Wan et al., 2007;Saiz et al., 2007; Balogh et al. 2011). That means there is high uncertainty on the relationship between soil respiration and temperature under different SWCs. It is necessary for us to use the data from our study area to check if the SWC plays an important role in the variation of soil respiration in our research area. Therefore, we choose two types of commonly used models (univariate and bivariate models), one about only soil temperature and the other containing soil temperature and SWC.
Reference:
Wan S., Norby R.J., Ledford J., Weltzin J.F., 2007. Responses of soil respiration to elevated CO2, air warming, and changing soil water availability in a model old-field grassland. Global Change Biology, 13: 2411-2424.
Saiz G., Black K., Reidy B., Lopez S., Farrell E.P., 2007. Assessment of soil CO2 efflux and its components using a process-based model in a young temperate forest site. Geoderma, 139: 79-89.
Balogh, J., Pintér, K., Fóti, S., Cserhalmi, D., Papp, M., & Nagy, Z. (2011). Dependence of soil respiration on soil moisture, clay content, soil organic matter, and CO2 uptake in dry grasslands. Soil Biology and Biochemistry, 43: 1006-1013.
6.Lines 269-270
the soil had a very low water content due to continuous high air temperature and scarce rainfall
According the climatic data, “annual precipitation of 1785 mm (line 83), annual evaporation averaged 1318 mm (line 86), your experimental plot can be characterized as humid. But according the lines 269-270 you state the need for addition soil moisture. In Introduction section you discuss the irrigation. It would be useful your assessment of the prospects of bamboo irrigation, and on our opinion your propositions in this way will become more weighable if you will use our new water use paradigm: https://plan.core-apps.com/acs_sd2019/abstract/6dc4fdd9-0f1c-41b3-a54e-ac4e18736726
Reply: Yes, the experimental plot was humid, but the distribution precipitation was uneven between different seasons, in the summer, bamboo irrigation was needed due to the high temperature, high evaporation and low precipitation, so we are appreciate your recommendation about your new water use paradigm and hope to use it in bamboo irrigation.
7.Lines 301-303
The interaction between the biotic factor and the artificial factor on the variation of monthly SR rate was 0.00%, and the interactions between the natural factor and the biotic and artificial factors were
0.31% and 0.07%,…
Statistical procedures authors used are attractive, and useful for experiment which conditions were strictly determined. But for the uncertain processes in natural conditions this methodology gives results which are at least strange. For some factors the procedure provides high accuracy calculation, but helplessly states the unexpected variation value of 55.40 %. If anyone will use the last result directly, it means that experiment was failed. Of course we do not think this way, but the fact mentioned, as well as a low correlation of studied factors needs addition explanation. The author’s expert opinion is needed to assess the used statistical procedures prospects, and possibilities to reveal the unknown factors of variation. It will help to make the future results more certain.
May be, the conclusion ought to contain the author’s proposition how to reduce the SR via additional pulse intra-soil watering.
One more important thing is a carbon dioxide use by plants as the way of carbon dioxide reversible sequestration, which helps to increase the volume of biological matter and the climate biological buffering. It would be useful to mention not only a standard assessment of carbon as a green house gas, but the fact that its green house capability is lower than danger for climate from the water vapor – most dangerous green house gas, which careful management, together with the biological matter management in focus of food, raw materials, bio-fuel production and climate stabilization are the real but very complicated tasks.
Reply: We appreciate your valuable suggestions and the results such as natural factor and the biotic and artificial factors were 0.31% and 0.07%, unexpected variation value was 55.40 %, we believe the reason is our work are rough, because of we only selected the mean temperature and soil water content as factor, there are more factors that affected soil respiration, and we need to precise our work in the future research. And we added “in the bamboo forests management we could reduce the SR by irrigation like additional pulse intra-soil watering.” in the conclusion section.

Reviewer 2 Report
Title: I suggest to make a statement with the main finding of the work, in order to be more attractive for the readers.
Introduction: please state the hypothesis of the work.
Line 76: “artificial” is inappropriate, please change it. Maybe “management” or "agronomy"? Please check the entire manuscript.
Figure 1: Is the air temperature the daily mean? Please state it. Please insert in the graph the standard error (or SD) in each value.
Lines 124-125: please state why you performed the measurement within 12:00-16:00.
Figure 3: please state what bars represent
Figure 5: I think this figure does not well represent the finding of the work because it put together factors on spatial variation (e.g. biomass) and on temporal variation (i.e. temperature and SWC). Could be better working on year cumulative values.
Figure 6: please state in caption what is repppresented by the bars
Line 370: please go more in depth. I suggest to see https://doi.org/10.1016/j.jaridenv.2019.02.008
Line 424: why annual? Just in the end of the monitoring period in figure 3.
Line 451: please go more in depth. I suggest to see https://doi.org/10.1016/j.catena.2016.12.013
Conclusions: please answer to the hypothesis to the work.
Author Response
Dear Sir,
Thank you very much for handling the review of our manuscript (forests-684464) entitled “Effects of soil temperature, water content, species, and fertilization on soil respiration in bamboo forest in subtropical China”. We appreciate the insightful comments and suggestions of anonymous reviewers as well as the subject editor, and have carefully considered each point brought up.
Based on the comments we received, careful modifications have been made to the original manuscript. Below you will find our point-by-point responses to the reviewer’s comments/questions.
Please note that: Reviewer’s comments/questions are in black. Our responses are in blue.
Sincerely yours,
Shunyao Zhuang, Ph.D.
1. Title: I suggest to make a statement with the main finding of the work, in order to be more attractive for the readers.
Reply: No good idea for the title. The main finding of this study is difficult to use one sentence to describe.
2. Introduction: please state the hypothesis of the work.
Reply: We appreciate your kind recommendation and we have added “we hypothesized that soil respiration of bamboo forests was primarily influenced by temperature, but it will be affected by other factors, such as soil moisture, management measures, bamboo species” in our manuscript.
3. Line 76: “artificial” is inappropriate, please change it. Maybe “management” or "agronomy"? Please check the entire manuscript.
Reply: We agree and we have changed the word “artificial” to “management” as suggested for the entire manuscript.
4. Figure 1: Is the air temperature the daily mean? Please state it. Please insert in the graph the standard error (or SD) in each value.
Reply: Yes, the air temperature was calculated based on daily measurement. We have added the standard error (SD) for each value.
5. Lines 124-125: please state why you performed the measurement within 12:00-16:00.
Reply: In order to choose the most representative time of the day for the measurement of SR as much as possible, we conducted a pre-experiment on a daily scale and made the measurement at intervals of two hours. The result indicates that the SR, soil temperature and water content showed similar trends in different treatments during one day. There are several hours at which the SR is close to the mean of the SR in one day which can be chosen as the representative time of the day for the measurement, such as 02:00-04:00, 12:00-16:00 and 20:00-22:00. Considering the night times at 02:00-04:00 and 20:00-22:00 at which it is not convenient to do the measurement, therefore we chose to perform the measurement within 12:00-16:00 at last. Although in some references the time for measurement would be chosen at 9:00-11:00, it is not the representative time in our study area
Figure 1 Soil respiration at different hours during one day (the data shows as Mean±SD)
Figure 2 Soil temperature and water content at different hours during one day (the data shows as Mean±SD)
6. Figure 3: please state what bars represent
Reply: The error bars represent SD of accumulative SR over the two years (June 2012-May 2014) for each treatment and the same character means there is no difference between the treatment. we have added the description in the manuscript of revised version.
7. Figure 5: I think this figure does not well represent the finding of the work because it put together factors on spatial variation (e.g. biomass) and on temporal variation (i.e. temperature and SWC). Could be better working on year cumulative values.
Reply: In order to analysis the quantitative contribution of different factors for the variation of SR, we try to select the most common factors and choose partial canonical correspondence analysis (PCCA) method which is commonly used in quantitative analysis. However, the PCCA method would not perform on the year-cumulative values due to the lack of measurement data (only one value of cumulative-SR for one year and two for the two-year monitoring period in total). Therefore, we conducted the PCCA based on the monthly data. However, we also discussed the difference of cumulative values of SR among different treatments and make some qualitative analysis about its factors in section 4.2 in the manuscript. We agree that the result would be better and persuading if the analysis could be performed based on year cumulative values. We really thanks for your suggestion and we will try such analysis in our future work.
8. Figure 6: please state in caption what is represented by the bars
Reply: The error bars represent SD of annual belowground biomass for each treatment. Different lowercase letters and uppercase letters indicate a significant difference for treatments in the year of June 2012 to May 2013 and in year of June 2013 to May 2014, respectively. We have added the description in the manuscript of revised version.
9. Line 370: please go more in depth. I suggest to see https://doi.org/10.1016/j.jaridenv.2019.02.008
Reply: We appreciate your kind recommendation and have added “Furthermore, some studies have suggested that the relationship between soil respiration and soil temperature varies according to moisture thresholds [26,27]” in our revised manuscript.
10.Line 424: why annual? Just in the end of the monitoring period in figure 3.
Reply: Although we have showed the cumulative soil respiration over the two year monitoring period in figure 3, what we focus on is the annual SR because we want to calculate the amount of carbon released from the soil to the atmosphere annually. Therefore, we mainly discussed the annual SR in this section.
11.Line 451: please go more in depth. I suggest to see https://doi.org/10.1016/j.catena.2016.12.013
Reply: We appreciate your kind recommendation and have added “the sensitivity of soil respiration to temperature significantly increased with C input from fertilizer [42].” in our revised manuscript.
12. Conclusions: please answer to the hypothesis to the work.
Reply: PCCA results showed that the natural factor (including ST and SWC) had a larger influence on the variation of the monthly SR rate in the experimental bamboo forest than the biotic (bamboo type) and management (fertilization) factors.
